# Relationship between Periodontal Condition of the Pregnant Woman with Preterm Birth and Low Birth Weight

**DOI:** 10.3390/jcm11226857

**Published:** 2022-11-21

**Authors:** Guillermo Gallagher-Cobos, Teresa Almerich-Torres, José María Montiel-Company, José Enrique Iranzo-Cortés, Carlos Bellot-Arcís, José Carmelo Ortolá-Siscar, José Manuel Almerich-Silla

**Affiliations:** 1Gynaecology and Obstetrics Service, Manises Hospital, 46940 Manises, Spain; 2Stomatology Department, University of Valencia, 46010 Valencia, Spain

**Keywords:** periodontal disease, preterm birth, low birth weight, risk factors, tobacco

## Abstract

The aim of this study was to determine the association between the mother’s periodontal condition and perinatal complications, such as preterm birth (PTB) and/or low birth weight (LBW), in a cohort of women in Valencia, Spain. Other related factors, such as tobacco, were also analysed. A prospective cohort study was carried out in a sample of 102 women with a single foetus and ages ranging between 18 and 42 years. Sociodemographic and obstetric variables, caries status, percentage of bleeding, clinical attachment loss (CAL), and probing pocket depth (PPD) data have been collected and analysed. The mean age was 32.4 years, and the BMI was normal. The average weight of new-borns (NB) was 3034 g. A total of 9.8% of the women smoked during their pregnancy. Bleeding percentage was 16.43% (SD 14.81%) and PPDs > 3 mm 8.8 (SD 11.08). The mean of CAL > 0 mm was 1.14 (SD 2.83). The frequency of PTB and LBW was 26%. No statistically significant differences were found between probing depth > 3 mm or CAL > 1 mm, with PTB and/or LBW. Periodontal disease in the mother was not statistically significantly related to either PTB or LBW. Tobacco use during pregnancy showed a statistical significance linked to LBW, but not with PTB.

## 1. Introduction

Preterm birth (PTB) is considered to be a birth occurring before the 37th week of gestation. There are two main distinct clinical subtypes of preterm birth: firstly, spontaneous preterm birth, which occurs naturally as a result of spontaneous onset of labour and/or premature rupture of foetal membranes; and secondly, premature birth indicated gynecologically due to a maternal or foetal pathology. Preterm labour is defined as regular contractions accompanied by a cervical change before the usual 37 weeks of gestation. It is initiated by multiple mechanisms, including infection or inflammation, cervical insufficiency, ischemia or uteroplacental haemorrhage, uterine distension, stress, and other immunologically mediated processes. In most cases, a single mechanism cannot be established. It is the synergy of multiple aetiologies and genetic–environmental interactions that can predispose a pregnant patient to PTB [1].

Preterm birth is one of the leading causes of neonatal mortality worldwide. Approximately 1 million preterm infants die each year because of complications stemming from preterm birth [2,3]. According to the World Health Organization (WHO), in the Born too Soon (2012) report, it is estimated that 15 million babies are born prematurely each year; this accounts for more than 1 in 10 births, and this figure is increasing. Of these premature babies, those who survive grow up with some type of disability, either of a cognitive nature, affecting their learning, or of a physical one; the latter is mainly related to the deterioration of lung function and development, as well as with sight and hearing. issues. In addition, they are more likely to develop medium- and long-term sequelae, such as cerebral palsy or developmental delays at a level ranging from 50% to 75% [4,5]. In the last two decades, knowledge about preterm birth has improved, however, its incidence has not decreased, which has spurred on the scientific community to investigate the associated risk factors [6].

The assessment of low birth weight (LBW) occurs when the baby’s weight is less than 2500 g; however, when the weight is below 1500 g, it is classified as a case of very low birth weight. Both circumstances pose a public health problem and are the main causes of most biological determinants of infant morbidity and mortality in developed and developing countries [7]. In most cases, there is an association between birth weight and weeks of gestation, although not always. Other factors can reduce intrauterine growth, such as multiple gestation, maternal pathologies, or infections [8].

Adverse pregnancy outcomes are also associated with the presence of elevated local and systemic inflammatory mediators and intrauterine infections. Current evidence suggests that adverse outcomes originate mainly from ascending vaginal or cervical infections; another source may be hematogenous spread from known or unknown non-genital sources. Maternal periodontitis represents a potential source of microorganisms that are known to routinely enter the circulatory system, and directly and/or indirectly can influence the health of the maternal–foetal unit [9]. Biological mechanisms that are likely to link periodontal disease to adverse pregnancy outcomes, such as PTB and LBW, appear to be related to inflammatory and immune responses and the suppression of local growth factors (IGF-2) in the fetoplacental unit. This mechanism could be articulated through a direct pathway, in which the spread of microorganisms would occur initially at the oral cavity or from the genitourinary tract, or take an indirect pathway, in which local inflammatory mediators produced in periodontal tissues (PGE2, TNFa) would affect the foetal–placental unit through the bloodstream; alternatively, they would circulate to the liver, triggering the production of cytokines (such as IL-6) that would act directly to have an impact on the foetal–placental complex [10].

The new classification of the AAP/EFP (American Academy of Periodontology and European Federation of Periodontology) aims to clearly identify the clinical entities of periodontitis by introducing operational elements, such as the detection of clinical attachment level (CAL), bleeding on probing (BOP), and probing pocket depth (PPD), to relate diagnosis to prevention and treatment needs. In the context of the 2017 World Workshop on the Classification of Periodontal and Peri-implant Diseases and Conditions, it was suggested that a single definition be adopted for the diagnosis of periodontitis [11].

The aim of this study has been to carry out a prospective study to analyse the relationship between the periodontal condition of the pregnant woman in the second trimester and perinatal complications, such as PTB and LBW, and analyse the relation of recognized risk factors, such as tobacco use.

## 2. Materials and Methods

A prospective cohort study was carried out in the Department of Obstetrics at Manises Hospital in Valencia, Spain. The sample size was determined by the prevalence of preterm births (<37 weeks) and the prevalence of births with low birth weight (<2500 g) based on published data for the same geographic area; the prevalence of preterm and low birth weight births among the Spanish population was 5.8% in 2010 [12]. A sample size of 84 women was deemed appropriate based on a confidence level of 95% and an accuracy of +/− 5 percentage unit. The pregnant women agreed to participate in the study by signing an informed consent form and the study protocol obtained authorization from the Research Ethics Committee of the La Fe University and Polytechnic Hospital in Valencia (reference number 2020-042-1).

We included 102 women who agreed to participate in the study and met the inclusion criteria of being between weeks 25 and 36 of gestation, with a single foetus, and aged between 18 and 42 years. Cases of multiple gestation, gestation of less than 34 weeks, and those over 37 weeks were excluded. Selection of the participants was made among all the pregnant women who attended their gynaecological pregnancy review in the Obstetrics Clinic of Manises Hospital.

For the oral examination, a dental unit, disposable gloves, and masks and a sterile dental kit composed of an intraoral flat mirror No. 5, bent tweezers, and a Williams periodontal probe were available. The examinations were conducted between December 2019 and February 2020, by a single explorer calibration. To ensure validity in the collection of data in fieldwork, the single explorer calibration (GGC) was performed with an experienced explorer that acted as gold standard (JMAS). The agreement was analysed using the intraclass correlation coefficient (ICC), which was 0.859 for all the measurements made, making it an almost perfect agreement according to Fleiss’ kappa. This agreement was not affected by the vestibular/palatine position of the tooth or by its location in the mouth (anterior or posterior), given that it maintained similar values in all positions.

The oral examination included a periodontal examination, in which probing depth (PPD) and clinical attachment loss (CAL) of all teeth present were determined and scanned in 6 locations: 3 buccal and 3 lingual or oral. Once the probing had been carried out, the number of bleeding points (BOP) was recorded and patients were classified as having or not having periodontitis according to the criteria of the 2017 World Workshop on the Classification of Periodontal and Peri-implant Diseases and Conditions. According to this framework, a patient is considered a case of periodontitis if there is a clinical presentation of one or more of the following: interdental CAL detectable at least 2 non-adjacent teeth, or buccal or oral CAL ≥ 3 mm with pocket depth greater than 3 mm detectable in at least 2 teeth, and that the CAL detected is not attributable to non-periodontal causes, such as gingival recession of traumatic origin, cervical caries that invade the root, CAL in distal of second molars associated with malposition or extraction of the third molar, an endodontic lesion that drains through the marginal periodontium, or a vertical root fracture [10].

A caries scan was also performed using the WHO diagnostic criteria to determine the DMFT index and caries prevalence among participants [13].

In addition, sociodemographic variables about the pregnant women were collected, such as age, level of completed schooling years or studies, medical variables (systemic, anthropometric diseases (body mass index, BMI)), habits (smoking), and obstetric history (previous pregnancies, vaginal births, Caesarean sections, or abortions).

In the second phase of the study, once the mothers had given birth, we proceeded to collect data related to childbirth—weeks of gestation and the weight of the baby, among others—to classify the birth as normal, low birth weight, or preterm birth. All data were collected in a pseudonymized form in a Microsoft Excel^®^ file.

A descriptive analysis was carried out using frequencies or percentages to describe qualitative variables and a bivariate analysis to establish the relationship between variables. For the comparison of proportions, the Chi-square test was used, along with the T-Student test or Mann–Whitney test to measure compliance with normality. For risk estimation, the odds ratio (OR) was used, and for all the above items, a significance level of 5% (*p* less than 0.05) was established, with a 95% confidence interval (CI). As a support for the statistical analysis, an Excel program was used, and the data were analysed with the SPSS programme 22.0, IBM^®^ (Armonk, NY, USA).

## 3. Results

### 3.1. Sample Description

In total, 102 women who attended their routine gynaecological examination between weeks 25 and 36 of pregnancy, met the inclusion criteria of the study, and agreed to participate were explored. The end sample was 98 women, because it was not possible to obtain delivery data for 4 women who gave birth in a different hospital (Figure 1).

In the initial assessment, a related to the condition of the mother, periodontal status, and caries status was carried out (Table 1).

Regarding the assessment of the level of completed years of schooling or education, a low level was considered to be the absence of studies or having completed only primary education, a middle or intermediate level was considered as having completed secondary education and/or the baccalaureate, while a high level was considered to be the completion of university studies or a master’s degree.

Regarding the periodontal condition, those mothers who, in the exploration, met the criteria established in the 2017 World Workshop on the Classification of Periodontal and Peri-implant Diseases and Conditions [10] were considered as periodontal disease patients. 

For caries status, the DMFT index of the sample presented moderate values (3.63), with a restoration rate of 65.3%.

### 3.2. Variables Related with the Childbirth

Regarding the variables related with childbirth, there were statistically significant differences in the presentation of preterm births with respect to the variables of new-born’s weight, length of time after rupturing of membranes, and assessment of the Apgar test at 5 min of life (Table 2).

### 3.3. Oral Health Mother Status Variables

Our results showed that neither the state of oral health nor the smoking habits of the mother seemed to significantly condition the presentation of PTB (Table 3).

However, the mother’s tobacco use did have a statistical significance in relation to low birth weight (Table 4).

The frequency of preterm birth was 26, as was that of low-birth-weight births, but not all women who had a preterm birth had low-birth-weight babies. These two circumstances only matched in 17 of the cases.

## 4. Discussion

This research has evaluated the relationship between the periodontal condition of the pregnant woman and preterm birth (PTB) and/or low birth weight (LBW) in a cohort of women from Valencia region. Some studies like ours have manifested a positive association between the periodontal condition of the mother with preterm births or neonates with low birth weight [14,15,16]. According to these authors, birth weight decreases with increasing probing pocket depth and clinical attachment level and, although this cannot always be attributed to periodontal disease of the mother, the severity of the mother’s periodontal disease during pregnancy has been related to a greater number of neonates with low birth weight. Similar results have been found in other similar studies [17,18,19,20]. Manrique-Corredor et al. [6] stated that maternal periodontitis doubled the risk of preterm birth and Offenbacher et al., in 1996, showed that periodontal disease is a significant risk factor for preterm low birth weight with an odds ratio of 7.9 [21]. De Oliveira et al. also recently published research showing that the mother’s periodontal diseases increase the risk of preterm birth, although this association is sensitive to the case definitions [22].

Other cohort studies, conducted with similar methodology to ours, have also shown significant association between mothers with periodontal disease during pregnancy and PTB and/or LBW [23,24,25,26,27]. However, in these studies, the sample size, the type of population, the age of the mother, and the collection of etiological or risk factors vary greatly, making it difficult to compare the results. This means that the currently available literature does not offer clear evidence on this relationship. The reasons for such heterogeneity are based on the type of examination, the calibration of the examiners, the stage of gestation at the time of examination, and the different definitions of periodontitis adopted as the cut-off point of the disease [28]. The meta-analysis by López et al. [1] that collected the results of eight prospective cohort studies where the association between periodontal infections and low birth weight and/or preterm birth was analysed, suggesting that periodontal disease does pose a risk for having preterm births or neonates with low birth weight.

While the definitions of preterm birth and low birth weight are well established, no clear consensus on the definition of periodontal disease (PD) has yet been reached in research, which is essential for optimizing the interpretation, comparison, and validation of clinical data [29]. The lack of uniform criteria for estimating periodontal disease in studies leads to disparate conclusions. For example, Offenbacher et al., in 1998 [30], defined PD as probing depth > 4 mm or average of CAL > 3 mm, while other researchers examined six points on all teeth that were present in the dental arch and considered periodontitis as those patients who had at least four teeth with one or more points with probing depth ≥ 4 mm and CAL ≥ 3 mm in the same point [31]. In 2005, Martins et al. [32] also agreed to use this definition for periodontitis and found a positive relationship between mothers diagnosed with periodontal disease and increased frequency of PTB or neonates with LBW. However, other research found no significant relationship between LBW and the mother’s periodontal disease considering these same criteria (probing depth ≥ 4 mm and CAL ≥ 3 mm) [33,34].

In our study, we have taken the definition of periodontitis proposed in the 2017 World Workshop on the Classification of Periodontal and Peri-implant Diseases and Conditions published by Tonetti et al. in 2018. According to this definition, a patient is a case of periodontitis if there is a clinical presentation of: interdental CAL in at least two non-adjacent teeth, or buccal or oral CAL ≥ 3 mm with probing pocket depth (PPD) greater than 3 mm detectable in at least two teeth and this CAL is not attributable to non-periodontal causes, such as gingival recession of traumatic origin, cervical caries that invade the root, CAL in distal zone of second molars associated with malposition or extraction of the third molar, an endodontic lesion that drains through the marginal periodontium, or a vertical root fracture. [10].

A meta-analysis published in 2013 [35] noted that the definitions of periodontitis commonly used by López et al. [1] resulted in statistically significant positive associations between maternal periodontitis and adverse pregnancy outcomes in the study population, but in contrast, analyses of data from the same populations using mean probing depths and other continuous variables frequently resulted in non-statistically significant associations. Other authors have also highlighted that periodontitis and periodontopathogens are not sufficient to trigger prematurity and/or low birth weight [36]. A case-control study investigating the association between periodontitis and adverse pregnancy outcomes, such as LBW, concluded that maternal periodontal disease cannot be considered as a risk factor for adverse pregnancy outcomes [37]. Other recent studies, such as that of Caneiro et al. from 2020 [29], conducted in a cohort of Spanish Caucasian women, concluded that moderate or mild periodontal disease is not consistent enough to trigger preterm birth; furthermore, Marquez et al. [38] pointed out that the prevalence of preterm births increased only in cases of severe periodontal disease. Studies by Bulut et al. [34], Moore et al. [39], and Srinivas et al. [40] could also not demonstrate such an association.

The presence of an association did not imply causality, although the results should always be interpreted with caution due to the allowance made for confounding variables and the high heterogeneity in the studies [41].

In addition to the mother’s periodontal disease, other risk factors related to prematurity or low birth weight have been studied, such as the mother’s age; the mother’s BMI; occupational exposure; tobacco, drug, and/or alcohol use during pregnancy; changes in the oral microbiome, such as oral dysbiosis [42]; the use of dental services during pregnancy [43]; short intergenic periods; having previous preterm delivery; use of assisted reproduction techniques; induced abortions [44]; and per capita income level, among other factors.

Pregnant women at high risk of preterm birth defined by obstetric criteria, such as twin pregnancy and history of previous preterm birth, but without periodontal disease, showed greater gingival inflammation compared to women with a normal course of pregnancy [45]. According to Reza et al. in 2015, in their study with groups of Iranian women where the majority were neither smokers nor alcohol users, the frequency of low birth weight in the new-borns of first-pregnancy women affected by periodontal disease was 2.3 times higher than that of those new-borns whose mothers had good periodontal health. However, the frequency of low birth weight in those new-borns of multiparous mothers affected by periodontal disease was six times higher than in the new-borns of women with periodontal health [46].

Smoking has been recognized as the most important risk factor for periodontitis. Smoker mothers have infants with lower birth weight than non-smoker mothers and, overall, smoker moms have higher infant mortality [47].

However, Manrique-Corredor et al. found in their systematic review that 20% of the studies did not take this risk factor into consideration [6]. Andonova et al. excluded pregnant smokers because they considered that it could be a factor leading to confusion [48]. Other studies [37,49] investigated variables such as smoking or alcohol consumption, and none of them were significantly associated with preterm birth or low birth weight. On the other hand, Resende et al. did mention a statistically significant association between a probing depth equal to or greater than 4 mm and smoking during pregnancy, considering this a relevant association because 20% of mothers of premature new-borns smoked during pregnancy [50]. In our study, smoking has been statistically significantly linked to low birth weight. Women smokers were 4.2 times more likely to have a baby with a low birth weight; however, the smoking habit has not been associated with preterm birth.

As for the relationship with the mother’s BMI, a low BMI at the beginning of pregnancy would be a risk factor associated with LBW and/or PTB; women with low weight gain during pregnancy, particularly those who started pregnancy with low BMI, showed an increased PTB risk. Women weighing less than 50 kg before delivery had 2.9 times more preterm births than women who went into labour at a normal weight and BMI [51]. It must be pointed out that not only can a low BMI affect the time of delivery, high BMI values are also related. Women with pre-pregnancy obesity and periodontitis are significantly more likely to have preterm birth with preeclampsia than pregnant women with periodontitis, but with normal weight [52]. However, other authors did not find this association [29,37]. Our study found no significant association between maternal BMI early in pregnancy and preterm birth or low birth weight.

Another factor related to the childbirth complications is the per capita income, level of the country or educational level of the mother. Moliner-Sánchez et al. [53] stated that the risk of having a preterm birth or a low-birth-weight baby in mothers with periodontal disease is increased in developing countries and less common in developed countries. In this study, the level of education of the mother was collected as an indicator of the child’s social level. In our sample, 45.1% had a university degree and/or master’s degree and were residents in Spain. These characteristics of the sample may have influenced the lack of relationship between periodontal disease in pregnant women and postnatal complications in this study.

### Limitations of the Study

The main limitation of our study would be the possible selection bias of the population, having limited the sample to pregnant women residing in the same city and belonging to the same hospital allocated to their area. However, the sociocultural level of the pregnant women in our study, obtained from the level of education of the mother, is representative of the sociocultural level of women at this time in Spain. The performance of oral examinations in different hospitals in different cities of our country, within the framework of a multicentric study, could undoubtedly increase the heterogeneity of the sample and, therefore, the external validity of the results.

Another limitation is that, in our study, it was only possible to perform an oral examination at a specific time (weeks 25–36 of pregnancy), given the organizational characteristics of the health centres where the samples were collected, other examinations at different stages of the pregnancy could have enriched the results. 

As we know, the periodontal condition can vary or be affected throughout pregnancy by hormonal changes, suggesting that, for future research, it would be advisable to carry out several measurements during pregnancy, including puerperal ones.

## 5. Conclusions

The relationship between periodontal disease of the pregnant woman and preterm birth and/or low birth weight has not been statistically significant in the population studied. Tobacco use during pregnancy has been associated with low birth weight in a statistically significant way. Female smokers are 4.2 times more likely to have a low-weight baby. However, the association of smoking with preterm birth has not been demonstrated. The mother’s body mass index in early pregnancy has not been significantly associated with either low birth weight or with preterm birth.

Regardless of the possible association or not, oral diseases should always be prevented, especially during pregnancy. It is the responsibility of health professionals seeing pregnant patients, such as dentists, obstetricians, gynaecologists, and midwives, to provide future mothers with information about the changes that will occur in their mouth during the course of the pregnancy; furthermore, the role of the said professionals is to make available the necessary preventive measures and treatments to minimize the risk inherent in pregnancy.

## Figures and Tables

**Figure 1 jcm-11-06857-f001:**
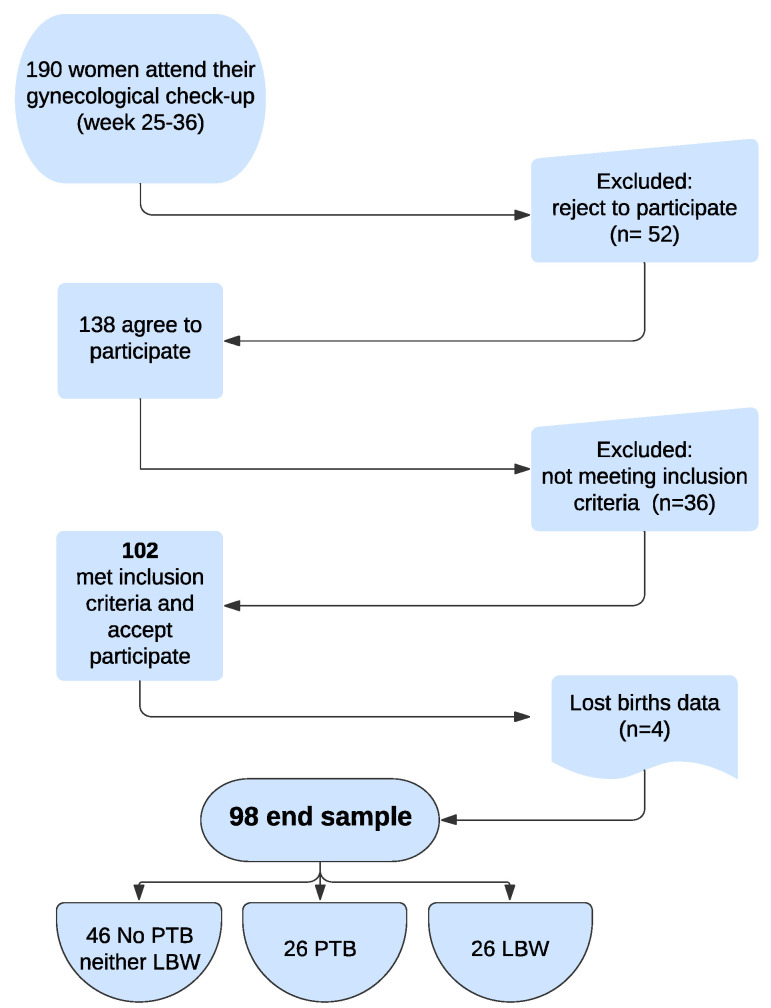
Flow chart about end sample selection.

**Table 1 jcm-11-06857-t001:** Characteristics of the sample.

Variable	Mean or % (CI at 95%)
Mother’s age	32.43 (31.32–33.53)
Initial Maternal BMI	23.11 (22.42–23.81)
Pregnancy history	1.11 (0.84–1.38)
Vaginal births	0.51 (0.33–0.69)
C-sections	0.13 (0.06–0.21)
Abortions	0.46 (0.33–0.59)
Level of education	
Low level	18.6% (12.3–27.3)
Middle level	36.3% (27.6–45.9)
High level	45.1% (35.8–54.8)
Periodontal condition	
Periodontal disease ^1^	24.5% (17.2–33.7)
% of bleeding	16.43% (13.52–19.35)
Pockets > 3 mm	8.83 (6.66–11.01)
Pockets > 5 mm	0.53 (0.23–0.83)
CAL > 0 mm	1.14 (0.58–1.69)
Caries status	
Component D (DMFT)	0.62 (0.34–0.90)
Component M (DMFT)	0.64 (0.37–0.91)
Component filled (BMFT)	2.37 (1.85–2.89)
DMFT	3.63 (2.97–4.28)
% caries-free (DMFT = 0)	15.7% (9.9% a 23.9%)
Prevalence of caries	84.3% (76.0–90.1%)

^1^ According to Tonetti 2018 criteria.

**Table 2 jcm-11-06857-t002:** Variables related with childbirth according to preterm birth (PTB).

Variable	Preterm Birth (PTB)		*p* Value Test
	YES (*n* = 26)	NO (*n* = 72)	
New-born’s weight (gr)	2450.8 (2278.9–2622.7)	3244.9 (3134.8–3355.2)	*p* < 0.001 *
New-born’s height (cm)	48.7 (48.4–49.1)	49.6 (48.4–50.7)	*p* = 0.396
Length of time after rupturing of membranes	4.31 (2.68–5.94)	6.86 (5.57–8.15)	*p* = 0.033 *
Apgar score at 5 min of life	8.96 (8.35–8.57)	9.51 (9.31–9.71)	*p* = 0.026 *
Arterial pH	7.17 (7.13–7.21)	7.19 (7.17–7.21)	*p* = 0.323
Epidural anaesthesia	80.8% (62.1–91.5)	94.4% (86.6–97.9)	*p* = 0.115
Local anaesthesia	3.8% (0.7–18.9)	1.4% (0.3–7.5)	
No anaesthesia	15.4% (6.2–33.5)	4.2% (1.4–11.5)	
Use of oxytocin	61.5% (42.5–77.6)	90.3% (81.2–95.2)	*p* = 0.001
Clear amniotic fluid	84.6% (66.5–93.4)	87.5% (77.8–93.3)	*p* = 0.710
Meconial amniotic fluid	15.4% (6.2–33.5)	12.5% (6.7–22.1)	
Spontaneous birth	57.7% (39.0–74.5)	66.7% (55.2–76.5)	*p* = 0.589
Instrumental vaginal birth	30.8% (16.5–50.0)	20.8% (13.1–31.6)	
C-section	11.5% (4.0–29.0)	12.5% (6.7–22.1)	

* Statistically significant.

**Table 3 jcm-11-06857-t003:** Oral health status according to preterm birth (PTB).

Variable	Preterm Birth (PTB)		*p* Value Test
	YES (*n* = 26)	NO (*n* = 72)	
% of Bleeding	17.0% (11.2–22.7)	16.4% (12.8–20.0)	*p =* 0.871
Pockets > 3 mm	10.81 (5.76–15.86)	8.51 (5.99–11.0)	*p =* 0.374
Prevalence of pockets > 3 mm	92.3% (75.9–97.9)	79.2% (68.4–87.0)	*p =* 0.129 OR = 3.2 (0.7–14.9)
Pockets > 5 mm	0.85 (0–1.69)	0.41 (0.11–0.73)	*p =* 0.233
Prevalence Pockets > 5 mm	23.1% (11.0–42.1)	16.7% (9.8–27.0)	*p =* 0.469 OR = 1.5 (0.49–4.5)
Points with CAL > 0 mm	1.88 (0.50–3.27)	0.90 (0.28–1.52)	*p =* 0.138
Prevalence of CAL	46.2% (28.8–64.5)	30.6% (21.1–42.0)	*p =* 0.152 OR = 1.94 (0.8–4.9)
Periodontal disease	30.8% (16.5–50.0)	23.6% (15.3–34.6)	*p* = 0.473 OR = 1.43 (0.53–3.88)
Component D (DMFT)	0.69 (0–1.39)	0.63 (0.31–0.94)	*p* = 0.841
Component M (DMFT)	0.46 (0.03–0.89)	0.74 (0.38–1.09)	*p =* 0.396
Component F (DMFT)	2.23 (1.31–3.15)	2.44 (1.78–3.11)	*p =* 0.731
DMFT index	3.38 (2.42–4.35)	3.81 (2.94–4.67)	*p =* 0.589
Prevalence of caries	84.6% (66.5–93.9)	84.7% (74.7–91.3)	*p =* 0.990 OR = 1 (0.3–3.4)
Smoker	17.4% (7.1–37.1)	7.2% (3.1–15.9)	*p =* 0.156 OR = 2.7 (0.7–11.0)

**Table 4 jcm-11-06857-t004:** Oral health status according to low birth weight (LBW).

Variable	Low Birth Weight (LBW)		*p* Value Test
	YES	NO	
% of bleeding	18.3% (13.0–23.6)	15.9% (12.2–19.6)	*p =* 0.871
Pockets > 3 mm	11.7 (11.5–16.8)	8.21 (5.73–10.7)	*p =* 0.374
Prevalence of pockets > 3 mm	84.6% (66.5–93.9)	81.9% (71.5–89.1)	*p =* 0.758 OR = 1.21 (0.4–4.1)
Pockets > 5 mm	0.85 (0–1.70)	0.42 (0.11–0.73)	*p =* 0.233
Prevalence Pockets > 5 mm	23.1% (11.3–42.0)	16.7% (9.8–26.9)	*p* = 0.469 OR = 1.5 (0.49–4.5)
Points with CAL > 0 mm	2.54 (0.48–4.59)	0.67 (0.40–0.94)	*p* = 0.138
Prevalence of CAL	46.2% (28.8–64.5)	30.6% (21.1–42.2)	*p* = 0.152 OR = 1.9 (0.8–4.9)
Periodontal disease	30.8% (16.6–50.0)	23.6% (15.3–34.6)	*p* = 0.473 OR = 1.43 (0.53–3.88)
Component D (DMFT)	0.85 (0.14–1.56)	0.57 (0.25–0.88)	*p* = 0.841
Component M (DMFT)	0.77 (0.24–1.30)	0.63 (0.28–0.97)	*p* = 0.396
Component F (DMFT)	2.35 (1.35–3.35)	2.40 (1.75–3.06)	*p =* 0.731
DMFT index	3.96 (2.78–5.14)	3.60 (2.76–4.43)	*p* = 0.589
Prevalence of caries	84.6% (66.5–93.9)	84.7% (74.7–91.3)	*p* = 0.990 OR = 1 (0.3–2.4)
Smoker	20.8% (9.2–40.5)	5.9% (2.3–14.2)	*p* = 0.034 * OR = 4.2 (1.02–17.3)

* Statistically significant.

## Data Availability

Not applicable.

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
