# Peer review of "Relationship between Periodontal Condition of the Pregnant Woman with Preterm Birth and Low Birth Weight"

_jcm, 2022, doi:10.3390/jcm11226857_

Round 1

Reviewer 1 Report

The subject of the study is interesting, however, they should continue the study and not just choose a single population in a city. A small sample can skew the results.

The study design is simple but sufficient and the main problem lies in the sample. This subject is discussed a lot, in many situations, always in the same way.

It will be necessary to vary factors and combine knowledge to show consistency.

I think that you should apply the questionnaire by increasing the sample and evaluating what you get, for example, through a multicentric study.

Author Response

We appreciate the reviewers’ comments after our first submission of this paper. All suggestions and comments have been considered point-by-point. The changes in the manuscript have been marked using the “Track Changes” function of MS Word.

Comments

The subject of the study is interesting, however, they should continue the study and not just choose a single population in a city. A small sample can skew the results.

The study design is simple but sufficient and the main problem lies in the sample. This subject is discussed a lot, in many situations, always in the same way.

It will be necessary to vary factors and combine knowledge to show consistency.

I think that you should apply the questionnaire by increasing the sample and evaluating what you get, for example, through a multicentric study

Response: The size of the study sample was adjusted based on the prevalence observed in a previous epidemiological study (see reference #12) and the level of education observed in future mothers (table 1) allows observing the distribution of the sample for comparison in future studies. We recognize that conducting a multicenter study is an interesting objective to address based on this study.

Reviewer 2 Report

The authors have written an interesting paper, which I enjoyed reading. My comments are the following:

1. The affiliation is confusing. Why do you have 7 affiliations, with no institution name but the same institutional email domain? You should insert the name and address for the Department of Obstetrics at Manises Hospital in Valencia, Spain and place all the email in a single line, after the full address. 

2. The first sentence of the abstract: "To determine the association between the mother’s periodontal condition and perinatal complications, such as preterm birth (PTB) and/or a low birth weight (LBW), in a cohort of women
in Valencia, Spain." does not have a subject. Please re-write it.

3. Minor English corrections are necessary (e.g.: "a patient is a case of periodontitis if ...")

4. Please include the date in the reference number of the authorization from the Research Ethics Committee of the La Fe University and Polytechnic Hospital in Valencia.

5. In you included pregnant women between 25-35 weeks of gestation (page 3, inclusion criteria), why did you exclude "gestation of less than 34 weeks"?

6. In your discussion about the study of Iranian women, could the "frequency of low birth weight in those newborns of multiparous mothers affected by periodontal disease" be produced by the de-mineralization of teeth because of multiparity (more children could lead to low calcium concentrations and thus to periodontal disease)? Have you considered this?

7. "Mothers with only primary education were classified as low level, medium
level was assigned those who completed high school and high level to those who obtained a university degree or master's degree" is repeated in the Discussion section (page 9) and should be removed from this section.

8. In the Limitations of the study section, the first sentence: "[..], hence other examinations at different stages." is abruptly cut. You need to end your sentence.

Great work!

Author Response

We appreciate the reviewers’ comments after our first submission of this paper. All suggestions and comments have been considered point-by-point. The changes in the manuscript have been marked using the “Track Changes” function of MS Word.

Responses:

The authors have written an interesting paper, which I enjoyed reading. My comments are the following:

  1. The affiliation is confusing. Why do you have 7 affiliations, with no institution name but the same institutional email domain? You should insert the name and address for the Department of Obstetrics at Manises Hospital in Valencia, Spain and place all the email in a single line, after the full address.

Response 1: The affiliation has been corrected and completed.

  1. The first sentence of the abstract: "To determine the association between the mother’s periodontal condition and perinatal complications, such as preterm birth (PTB) and/or a low birth weight (LBW), in a cohort of women in Valencia, Spain." does not have a subject. Please re-write it.

Response 2: The sentence has been completed. Thank you for your comment.

  1. Minor English corrections are necessary (e.g.: "a patient is a case of periodontitis if ...")

Response 3: This sentence has been corrected. Thank you for your comment.

  1. Please include the date in the reference number of the authorization from the Research Ethics Committee of the La Fe University and Polytechnic Hospital in Valencia.

Response 4: The authorization of the study protocol was obtained on November 20, 2019 and the final resolution of the authorization (with the reference number) of the Research Ethics Committee of the La Fe University and Polytechnic Hospital in Valencia, was obtained on April 22, 2020.

  1. In you included pregnant women between 25-35 weeks of gestation (page 3, inclusion criteria), why did you exclude "gestation of less than 34 weeks"?

Response 5: The Department of Obstetrics of the Manises Hospital does not attend very premature births (below 34 weeks of gestation)

  1. In your discussion about the study of Iranian women, could the "frequency of low birth weight in those newborns of multiparous mothers affected by periodontal disease" be produced by the de-mineralization of teeth because of multiparity (more children could lead to low calcium concentrations and thus to periodontal disease)? Have you considered this?

Response 6: the decrease in blood calcium levels is not related to the possible demineralization of teeth in adults. This dental demineralization is more related to the intake of sugars, salivary factors or oral hygiene. In any case, this dental demineralization is related to the presence of caries but not to the presence of periodontal disease, which is the objective of this work.

  1. "Mothers with only primary education were classified as low level, medium

level was assigned those who completed high school and high level to those who obtained a university degree or master's degree" is repeated in the Discussion section (page 9) and should be removed from this section.

Response 7: It has been removed. Thank you for your comment.

  1. In the Limitations of the study section, the first sentence: "[..], hence other examinations at different stages." is abruptly cut. You need to end your sentence.

Response 8: The sentence has been completed. Thank you for your comment.

Great work!

Response: Thank you

Reviewer 3 Report

The study, Relationship Between Periodontal Condition of the Pregnant Woman with Preterm Birth and Low Birth Weight, has investigated how is the oral status (PD and caries) associated with PTB and LBW. The goal of this study is reasonable, each step of the study is sound and described with great details. However, there are not too much “statistically significant data”, but this is acceptable since we should also publish “negative data” to support an unbiased scientific view. But there are still some minor issues that the authors need to address to. 

1.     Abstract. Typo. The LWB in the abstract should be changed to LBW.

2.     The including and excluding criteria. Should pregnant women with systemic diseases be excluded? Please specify.

3.     Just wondering, do the authors have the record of infant’s medication treatment or hospitalization after a certain time (i.e. 3 months or 6 months or 1 years) of the birth? If they have it, they may also carry out some analysis. 

Author Response

Comments:

The study, Relationship Between Periodontal Condition of the Pregnant Woman with Preterm Birth and Low Birth Weight, has investigated how is the oral status (PD and caries) associated with PTB and LBW. The goal of this study is reasonable, each step of the study is sound and described with great details. However, there are not too much “statistically significant data”, but this is acceptable since we should also publish “negative data” to support an unbiased scientific view. But there are still some minor issues that the authors need to address to.

Responses: 

We appreciate the reviewers’ comments after our first submission of this paper. All suggestions and comments have been considered point-by-point. The changes in the manuscript have been marked using the “Track Changes” function of MS Word.

  1. Abstract. Typo. The LWB in the abstract should be changed to LBW.

Response 1: It has been changed. Thank you for your comment.

  1. The including and excluding criteria. Should pregnant women with systemic diseases be excluded? Please specify.

Response 2: The aim of this study was to determine if there is an association between the periodontal condition of pregnant women and preterm birth or low birth weight. The systemic diseases of the women, as well as their medications were collected, but in our study, we not excluded women because they had a systemic disease.

  1. Just wondering, do the authors have the record of infant’s medication treatment or hospitalization after a certain time (i.e. 3 months or 6 months or 1 years) of the birth? If they have it, they may also carry out some analysis.

Response 3: Analyzing the data longitudinally after childbirth was not one of the objectives of the study, which is why these data have not been collected. We appreciate the suggestion for a new study.